# HL-IR mediates cinnamaldehyde repellency behavior in parthenogenetic *Haemaphysalis longicornis*

**Ceyan Kuang, Han Shi, Jie Cao, Yongzhi Zhou, Houshuang Zhang, Yanan Wang, Jinlin Zhou**⬤*

Key Laboratory of Animal Parasitology of Ministry of Agriculture, Shanghai Veterinary Research Institute, Chinese Academy of Agricultural Sciences, Shanghai, China

* jinlinzhou@shvri.ac.cn

## Abstract

Chemical repellents against arthropods have limitations in terms of toxicity and resistance. Natural plant compounds can be utilized as alternatives for developing environmentally friendly repellents for humans and animals. A variety of plant essential oils exhibit strong repellent effects against ticks; however, the mechanisms of action against ticks remain unknown. Here, we investigated the repellency of cinnamaldehyde, a primary compound found in cinnamon oil, and demonstrated that it affected the electrophysiological responses on Haller's organs of parthenogenetic *Haemaphysalis longicornis*. Transcriptome data indicated that the cinnamaldehyde response was linked to ionotropic receptor (HL-IR) at various tick developmental stages. HL-IR was widely expressed in a variety of tissues and developmental stages of ticks according to RT-qPCR. In situ hybridization results showed that HL-IR was highly expressed on Haller's organs of the ticks. Microinjection of HL-IR double-stranded RNA (dsRNA) showed that reduced transcript levels led to significant decreases in the tick repellency rate from cinnamaldehyde and the EAG response of Haller's organ. Experiments using competitive fluorescence binding and mutation sites showed that 218ASN was the critical binding site for cinnamaldehyde and HL-IR. We conclude that Haller's organ of ticks expresses HL-IR, and that this interaction mediates tick-repellent behavior by binding to cinnamaldehyde.

## Author summary

Tick and tick-borne diseases present a significant threat to human and animal health. To mitigate tick bites and the spread of tick-borne diseases, chemical repellents are commonly utilized in daily life. In recent years, the resistance and toxicity of chemical repellents have been brought to light, leading to a growing interest in plant-derived repellents as a new research focus. Ticks sense their environment primarily through their sense of smell because their vision and hearing are underdeveloped. Despite this, the mechanisms underlying tick olfaction remain poorly understood. The interaction between ticks and plant-derived chemicals is a fascinating area of study. Our findings revealed Haller's organ is the key sensory organ for tick sensing of cinnamaldehyde. We reveal an ionotropic

**Data availability statement:** The authors declare that the data for the results of this study are available in the paper and its Supplementary Information. All relevant raw data provided in the main image and supplementary information have been made available in the source data files. The raw reads of our transcriptome data have been deposited into the NCBI Short Read Archive (SRA, http://www.ncbi.nlm.nih.gov/sra/) under accession number PRJNA1133957.

**Funding:** The National Key Research and Development Program of China (2022YFE0120100 to JZ). The funders had no role in study design, data collection and analysis, decision to publish, or preparation of the manuscript.

receptor in ticks that specifically detects cinnamaldehyde, triggering repellent responses through Haller's organs, and elucidated its binding site. A comprehensive understanding of this process is imperative for effective tick prevention and disease control.

## Introduction

Arthropods are the world's most diverse animal taxa, and many species act as vectors for a wide range of diseases, posing a substantial threat to human and animal health [1–3]. Arthropod survival and reproduction may depend on the capacity to detect natural odors and respond to changing odor concentrations [4–6]. Chemical repellents have been utilized to effectively prevent and control vector arthropods and the diseases they transmit [7]. However, the widespread usage of chemical repellents can be harmful to humans and animals. At the same time, repellent resistance is becoming a serious problem [8–10]. Arthropods and plants often cohabit in a mutually beneficial relationship [11,12]. Natural repellents not only help to maintain ecosystem homeostasis but also provide an alternate method of controlling arthropods [13–15]. Hundreds of essential oils have been demonstrated to possess anti-tick properties, which from the *Liliaceae*, *Araucariaceae*, *Boraginaceae*, *yrtaceae*, *Cupressaceae*, *Asteraceae*, and so on [16–20]. Recent research has indicated a significant repellent effect against ticks in Chinese cinnamon oil[21,22], In our previous study, we found that cinnamaldehyde is the main component of cinnamon essential oil[23].

Ticks are the second most important arthropod disease vector after mosquitos. There are over 900 species of ticks in the world. *Haemaphysalis longicornis* is a common species in China [24,25]. Ticks can induce allergic reactions, anemia, fatigue, and malnutrition in cattle as well as major health concerns in animal products (e.g., wool, meat, and milk) [26]. The spread of ticks and tick-borne diseases is a significant public health challenge [27,28]. Due to their underdeveloped visual and auditory senses, ticks primarily rely on their olfactory sense, which also plays a major role in host selection. Ticks' unique olfactory sensory structure is known as Haller's organ, a structure that can detect a wide variety of scents and transmit signals to the brain-like structure, the synganglion, for neural signaling and processing. Haller's organ, situated on the tarsal segment of the tick's first pair of legs, is recognized as a crucial sensory organ for ticks. Haller's organ of ticks can detect a variety of volatile substances such as acids, ketones, phenols, and esters [29–32]. The genomic analysis of ticks has identified a wide range of chemosensory proteins, but their roles remain unknown [33].

The primary excitatory neurotransmitter, glutamate, plays a crucial role in learning, behavior, emotion, and neural communication[34,35]. In both eukaryotes and prokaryotes, glutamate ionotropic receptors (iGluRs) participate in neurotransmission and signaling pathways in response to external chemical stimuli [36]. The ionotropic receptors (IRs) in arthropods are cation-gated channels ($Na^+$, $K^+$, or $Ca^{2+}$) derived from the common family of ionophilic glutamate receptors and synaptic ligand-gated ion channels. These are linked to a range of chemosensory sensations [37]. A total of 15 IRs have been screened in the transcriptomics of *Ixodes scapularis*, while 43 IRs have been identified in the genomics of *H.longicornis*. However, to date, no functional validation of tick IRs has been accomplished[33,38].

This study investigated the association between tick olfaction and cinnamaldehyde, a primary compound found in cinnamon oil. We examined the action mechanism of cinnamaldehyde against ticks.

## Materials and Methods

### Ethics Statement

Animals were maintained in a sterile atmosphere at the Animal Center of the Shanghai Veterinary Research Institute. All animal experiments were authorized by the Animal Ethical Committee of Shanghai Veterinary Research Institute (approval number SV-20211117-04, SV-20210910-02).

### Ticks, cells, and chemical reagents

A colony of *H. longicornis* ticks (parthenogenetic colonies) was collected from the Wildlife Park of Shanghai, China. *H. longicornis* were fed on New Zealand rabbits during the life cycle and were kept in the laboratory at 25°C and 90% humidity according to standard rearing procedures [39,40]. Ticks that had molted for ten days were collected as test material. All ticks were collected after feeding blood on rabbits, one batch of ticks was used for each experiment. HEK 293T cells were cultured at 37°C with 5% $CO_2$ in DMEM (Gibco, USA), supplemented with 10% heat-inactivated fetal bovine serum (Gibco, USA) and 2% Penicillin-Streptomycin (Beyotime Biotechnology, China). The following reagents were employed: cinnamaldehyde (Sinopharm Chemical Reagents, China), DEET (Macklin, China), Icaridin and IR3535 (yuanye, China).

### Behavioral assay

Referring to our previous repellent test, the repellent effect of odorants was measured using a Y-tube olfactometer [23]. Samples were obtained from unfed developmental stages of ticks at 7–15 d after hatched. The Y-tubes were perforated at the top and covered with a breathable mesh cloth. 80 female adults, 100 nymphs, or 300 larvae were positioned at the bottom of the tube. Each tube contained a filter paper containing 10 μL of test solution, which is 95% ethanol dilution of 0.25%, 0.5%, 1%, 2%, 4% of cinnamaldehyde or 20% DEET, and the other one had a filter paper containing 10 μL of 95% ethanol (Sinopharm chemical reagents, China) as a control. To ensure free movement of the ticks, the tubes were placed vertically on a tabletop. After 360 min, the numbers of ticks at each end were observed and counted at 8 time point. For statistical analysis, 6–8 independent trials were performed. Dunnett's Multiple Comparison-test was used for 360 min, student's t-test was used for 120 min time point analysis, and the data were plotted using GraphPad Prism 5. The repellency rate was then calculated using the formula:

$$\text{Repellency} \left( \% \right) = \frac{\text{number of ticks on 95\% ethanol - number of ticks on the odor source}}{\text{number of ticks on 95\% ethanol}} \times 100\%$$

### Scanning electron microscope observation of Haller's organ

The first legs of ticks at different developmental stages (larvae, nymphs, and female adults) were taken and washed three times in PBS (Servicebio, China) and then put into the electron microscope fixative for 2 h at room temperature then store at 4°C; after fixation, the samples were put into 0.1 M PB (0.1 M phosphate buffer pH 7.4) (Servicebio, China) and rinsed three times for 15 min each time. The cleaned samples were fixed in 1% osmium acid (0.1 M PB preparation) (Ted Pella, USA) for 90 min at room temperature and protected from the light, and then rinsed in 0.1 M PB three times for 15 min each time. The tissues were dehydrated in a sequence of alcohol concentrations (30%–50%–70%–80%–90%–95%–100%–100%)

(Sinopharm Chemical Reagent, China) for 15 min each time. After soaking in isoamyl acetate (Sinopharm Chemical Reagent, China) for 15 min, the samples were put into a desiccator (Quorum, UK) for drying; finally, the samples were fixed on conductive carbon film double-sided adhesive and sprayed with gold for 25 s (HITACHI, Japan). The samples were observed under a scanning electron microscope (HITACHI, Japan).

## EAG Recordings

A suitable amount of conductive adhesive was applied to a fork, and the first leg of the tick was cut off and quickly fixed onto a conductive adhesive, and the Haller's organ was exposed to air before applying a suitable amount of conductive adhesive to prevent drying. The Haller's organ was kept under a constant flow of purified and humidified air (170 mL/min). The odorants to be measured were diluted with paraffin oil (Servicebio, China). Then, 10 µl of each odorant solution was loaded on a 5 cm × 2.5 cm filter paper and placed in a Pasteur pipette, and the Haller's organ were exposed to the stimulus for 0.6 s, with 8-s intervals between administrations. Each test used a minimum 30 s interval between stimuli. The EAG responses were recorded using IDAC-2 (Syntech, China) and analyzed by EAGpro software. Each test was repeated 6–8 times.

## Transcriptome library construction and analysis of 2% cinnamaldehyde stimulation

Unfed ticks were divided into two groups (100 female adults, 200 nymphs, or 600 larvae); 10 µl of diluted 2% cinnamaldehyde was dropped onto a 1-cm-diameter filter paper, while 10 µl of 95% ethanol was added to the control group, 10 µl of 2% cinnamaldehyde (95% ethanol dilution) was added to the treat group. The filter paper was placed in the tubes for 30 min (larvae 15 min), and then the tested tick was frozen with liquid nitrogen. Then, 1 mL of Trizol (Invitrogen, USA) was added to mechanically homogenized the ticks, and total RNA was extracted. Two micrograms of total RNA was employed for transcriptome library construction (Beijing Genomics institution, China), and then sent to genedenovo for analysis. The extracted mRNA was enriched and then reverse-transcribed into DNA using N6 primer and synthesized into double-stranded DNA. After forming sticky ends, specific primers were used for PCR amplification, and the DNA was converted into cyclic DNA using a bridge primer and then sequenced on the DNBSEQ platform. Data from transcriptome sequencing were charged; reads were assembled and assessed for completeness using Trinity software, and Unigenes were annotated via the KEGG (http://www.geneontology.org/), GO (http://www.geneontology.org/), COG (http://www.ncbi.nlm.nih.gov/COG/), Swiss-Prot (http://web.expasy.org/docs/swiss-prot_guideline.html), Pfam (http://pfam.xfam.org/), and NR (http://www.ncbi.nlm.nih.gov/blast/db/) databases. Genes with FDR < 0.05 and |log2FC|>1 were screened for significant differences.

## Cloning and sequence analysis of HL-IR

After analyzing the transcriptome results in conjunction with the published genome[33] of the tick, we selected HL-IR for cloning, and total RNA samples were extracted from 30 ticks using TRIzol reagent (Invitrogen, USA). The cDNA was synthesized from 1 µg of total RNA using a cDNA synthesis kit according to the manufacturer's instructions (Takara, Japan). The cDNA was amplified by PCR using primer prime 6.0 software to design specific primers (S1 Table). The reactions were run under the following conditions: 98°C for 1 min, 30 cycles of 98°C for 10 s, 57°C for 15 s, 72°C for 30 s, and 72°C for 7 min. The PCR products were analyzed by 1% agarose gel electrophoresis and ligated onto the pMD-19T vector (Vazyme, China). Positive

strains were taken for sequencing (saihengbio, China). Positive results were selected for structural domain prediction using NCBI, and sequence comparison and phylogenetic tree construction in MEGA11 were performed using databases for different arthropods.

## Dynamics of tissue expression and cinnamaldehyde stimulation of HL-IR by qRT-PCR

To analyze the mechanism by which ticks responded to cinnamaldehyde, qRT-PCR assays were carried out for the transcripts of HL-IR. Totals of 400 larvae, 80 nymphs, and 60 female adults in the unfed stage were divided into two groups and presented with 10 μL drops of 95% ethanol or 2% cinnamaldehyde (diluted in 95% ethanol) in 1-cm pieces of filter paper. The samples were removed after they had been left in the tick tubes for 30 min, and the experiment was repeated three times for each group.

To detect the transcript levels of HL-IR in various tissues and at several developmental stages, about 600 female adults were dissected using a stereomicroscope for qRT-PCR to collect tissues, including four pairs of legs, ganglia, midgut, ovaries, salivary glands, and fat bodies. The experiment was repeated three times for each group.

After collecting the samples, total RNA was extracted, and cDNA was synthesized using a HiScript III RT SuperMix for qPCR (+gDNA wiper) kit (Vazyme, China) for RT-qPCR according to the manufacturer's instructions. Specific primers are listed in S1 Table. A house-keeping gene (Elongation Factors IA, ELFIA) was used as a reference. The qRT-PCR was carried out by using ChamQ Universal SYBR qPCR Master Mix manufacturer instructions (Vazyme, China). QuantStudio Q5 quantitative PCR instrument (Applied Biosystems, USA) was used to detect the Ct values, which were calculated by the $2^{-\Delta\Delta Ct}$ method. Student's *t*-test was used for analysis, and the data were plotted using GraphPad Prism 5.

## Construction, mutation, and protein purification.

Primers were designed (S1 Table) and the target fragment was cloned into a pET-30a vector (Takara, Japan). The positive bacterial solution was amplified and subjected to IPTG (BIOF-ROXX, China) induction at 25°C for 16 h. The bacterial solution was fragmented using ultra-sonication, and then the proteins were purified using BeaverBeads IDA-Nickel (Beaverbio, China) according to the manufacturer's instructions.

For the mutation experiment, a Mut Express II Fast Mutagenesis Kit V2 (Vazyme, China) was used, with primers designed to mutate HL-IR (S1 Table), and the mutated plasmid was used for protein purification following the above approach.

## Antibody Preparation and Expression Level Detection of HL-IR by Western Blotting

The B cell linear epitopes of HL-IR proteins were predicted using the Immune Epitope Database (IEDB) (http://tools.iedb.org/bcell/result/). The peptides (S1 Table) were synthesized by GL Biochem (China). Balb/c mice were intraperitoneally injected with an emulsified mixture containing an equivalent amount of Freund's adjuvant (Sigma-Aldrich, USA). A total of four injections were administered at two-week intervals, and blood was collected for Western blotting.

Following the separation of total proteins on 10% SDS-PAGE gels, the proteins were transferred onto PVDF membranes (Millipore, USA). Protein extracts from unfed nymphs were detected using the sera anti-His-HL-IR, whereas the signal from the target protein was normalized using an anti-α-tubulin primary antibody (Proteintech, USA) as a constitutive control. Secondary antibodies were obtained using goat anti-mouse IgG conjugated with HRP

(Invitrogen, USA). Image analysis was performed using an automated chemiluminescence system (Tanon Science & Technology. China).

## Identification of HL-IR by *in situ* hybridization

Hybridization sequences were designed, and luciferase sequences were used as a negative control (S1 Table). The first legs of female adults were fixed and sliced into paraffin sections. After dewaxing, the sections were digested with proteinase K (20 ug/mL) at 37° for 10 min and washed three times with PBS. Pre-hybridization was performed by incubation at 37°C for 1 h; a 500-nM probe was added, and the samples were hybridized overnight at 42°C in a humid chamber, then washed with SSC three times for 10 min each time. The slides were exposed to buffer for 45 min at 40°C, then washed three times with SSC for 5 min each time. Drops of signal hybridization solution containing Anti-DIG antibody were added and incubated for 45 min at 40°C in a humid chamber, then washed with SSC three times for 5 min each time. DAPI staining was performed for 8 min in the dark, and the slices were sealed with anti-fluorescence quenching sealer. The slices were sealed and then observed under an orthogonal fluorescence microscope, and images were captured for development. The nuclei stained by DAPI were blue under ultraviolet excitation, and the positive expression was a type of fluorescence labeled by the corresponding luciferin, where cy3 indicates red light. The reagents were obtained from Servicebio, China.

## Subcellular localization of HL-IR

The target fragment was cloned into an EGFP-N1 plasmid (Addgene, USA). Primers list in S1 Table. Positive bacteria were amplified, and the plasmid was extracted and transferred into HEK 293T cells (CTCC, China) in the subtitle. The 2μg plasmid transfection was carried out by using lipo3000 (Invitrogen, USA). Cells were washed after 24 h, fixed in ethanol, and washed three times with PBS, incubated for 5 min in 0.2% Triton X-100, then rinsed with PBS three times and incubated for 1 h at room temperature. The samples were incubated with $Na^+$, $K^+$-ATPase α1 (positive control) (1:50 dilution, CST, USA) and GFP tag Monoclonal antibody (1:25 dilution, protrintech, USA) as primary antibody overnight at 4°C, then washed three times with TBST (0.1% Tween diluted in Tris Buffered Saline) for 10 min each time. Alexa 488-labeled goat anti-rabbit antibody and Alexa 594-labeled goat anti-mouse antibody (Invitrogen, USA) were incubated with secondary antibody at room temperature for 1.5 h, washed three times with TBST for 10 min each time, then incubated with DAPI (Hoechst, Invitrogen, USA) for 15 min. The slices were sealed and examined under a ZEISS laser confocal microscope (German).

## RNA interference analysis of HL-IR

The double-stranded RNA (dsRNA) was synthesized *in vitro* using a T7 RiboMAX Express RNAi System kit (Promega, USA). The HL-IR dsRNA fragment contained 519 bp, and the luciferase dsRNA contained 573 bp. The synthesized dsRNA was stored at -80°C.

For knockdown of HL-IR in the nymphal stage, 9.2 nl of 1500 ng/μl dsRNA was injected at the base of the fourth pair of legs of each tick, and then the ticks were placed in clean tubes and stored at room temperature under high humidity for about 48 h.

## Fluorescent binding of HL-IR

N-phenyl-1-naphthylamine (1-NPN) is a commonly used fluorescent probe, and the effect of recombinant protein binding to 1-NPN is determined by the measurement of fluorescence. HL-IR protein solution was diluted to 2 μM in 50 mM Tris-Cl buffer (Sinopharm chemical

reagents, China), and placed in a 1-cm light path quartz cuvette; the fluorescent probe, 1-NPN (Meilunbio, China), was diluted to 1 mM in methanol (Sinopharm chemical reagents, China). The fluorescence emission spectrum from 350 to 550 nm was obtained by excitation at 337 nm.

For the dissociation constant ($K_d$) assay, HL-IR protein solution was titrated with a 1-NPN gradient to a final concentration of 20 μM, and the values of each fluorescence emission spectrum were measured. The value of $K_d$ was calculated from a Scatchard plot of the binding value, and the assay was performed in three replicates.

The fluorescence competitive binding assay was as follows: 1-NPN and protein were both configured to 2 mM, and a methanol solution of 1 mM odor molecules was titrated to a final concentration of 40 mM. The fluorescence intensity for each titration was recorded. Dissociation constants of competing ligands were calculated using the IC50 value. The apparent binding affinity ($K_i$) for each odorant was calculated using the Scatchard equation, plotted by Prism software, and each set of experiments was repeated three times. Ki was calculated according to the following equation:

$$K_i = [IC50] / \left(1 + [1\text{-NPN}]/K_d\right)$$

where [IC50] is the concentration of each ligand added when the fluorescence intensity of 1-NPN was reduced to 50%, and [1–NPN] is the free concentration of 1–NPN.

### Prediction of the binding site of HL-IR to cinnamaldehyde

The crystal structure of HL-IR protein was obtained using SWISS-MODEL (Expasy, CH); the protein and cinnamaldehyde were preprocessed using Schrödinger's software (Schrödinger, USA) as follows: the protein was processed using the Protein Preparation Wizard module, and the LigPrep module was used to generate the 3D structure of the cinnamaldehyde chiral molecule; the binding sites were predicted using Schrödinger's SiteMap. Receptor Grid Generation and Enclosing box modules of Schrödinger's software were used to predict the active sites. Molecular docking was then carried out; finally, MM-GBSA was used to determine the active sites.

## Results

### Ticks' responses to cinnamaldehyde via Haller's organ.

A Y-tube choice assay was conducted to test the chemosensory response of *H. longicornis* ticks to cinnamaldehyde. Behavioral assays at different developmental stages for 6 h showed that the nymphal stage was the most sensitive to cinnamaldehyde (S1 Fig). Concentration gradient behavioral tests revealed that 2% cinnamaldehyde had a repellent effect on all three developmental stages. Here, we have chosen to present the results for 120 min. The repellency percentages of cinnamaldehyde against larvae, nymphs, and female adults were 99.17%, 100%, and 80.67%, respectively. (****$p < 0.0001$). Compared to the negative control (95% ethanol), the ticks in different developmental stages were repelled by 2% cinnamaldehyde within 120 min, with nymphs being the most sensitive (Fig 1A). There was no significant difference between the positive control (20% DEET) and the cinnamaldehyde group in any of the developmental stages.

Haller's organ, like the antennae of insects, displays distinct sensitivity to environmental cues. We observed that Haller's organ had an adequate structure at different developmental stages of *H. longicornis.* The larvae differ from the nymphs and female adults in that they have one less sensory hair (Fig 1B). Within two hours of removing Haller's organ, both the nymphs

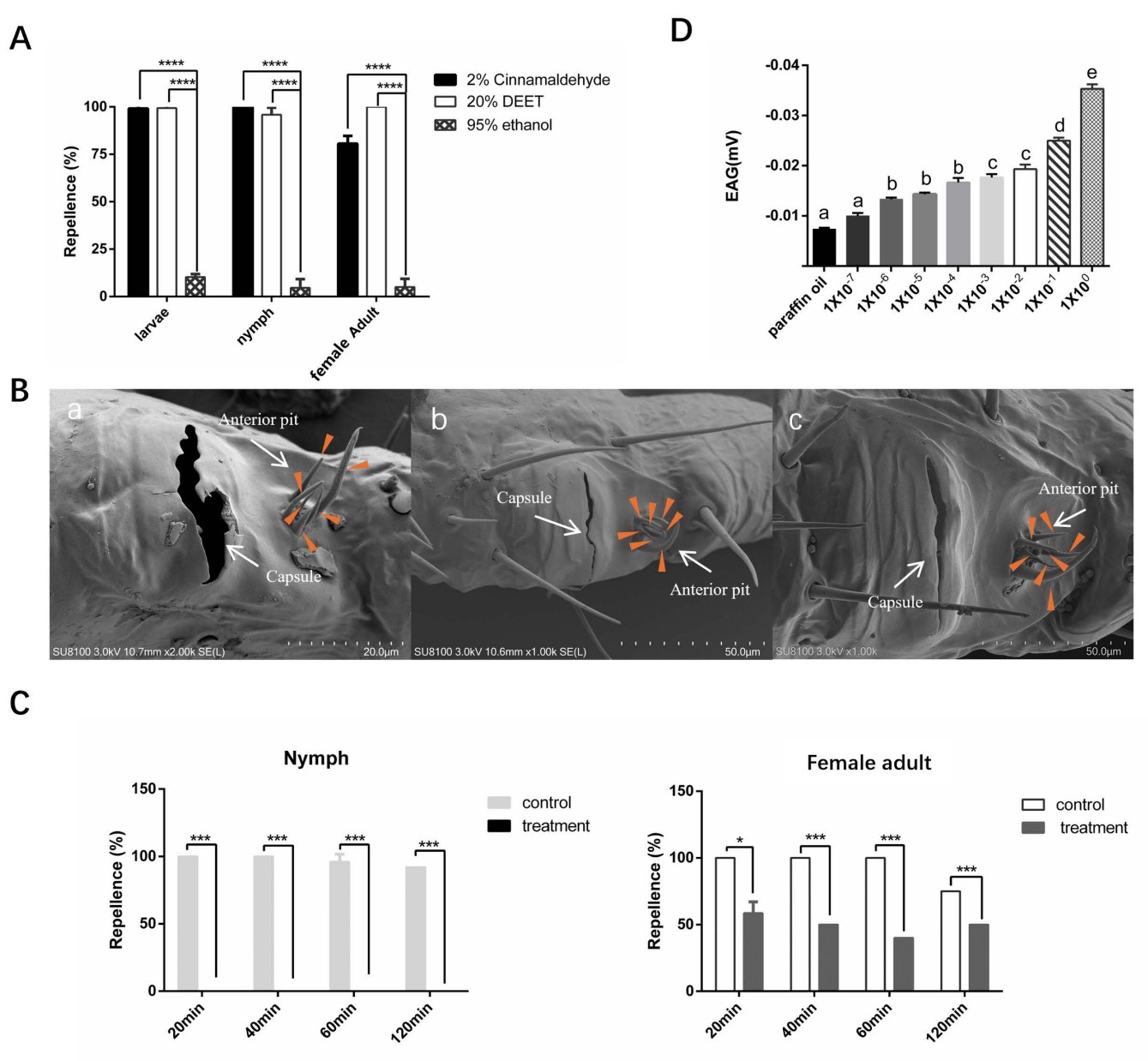

**Fig 1. Haller's organ is the key sensor for cinnamaldehyde repellency in ticks.** A. Cinnamaldehyde repellency test within two hours in different stages of *H. longicornis*. The repellency percentages of cinnamaldehyde against larvae, nymphs, and female adults were 99.17%, 100%, and 80.67%, respectively. Samples were obtained from unfed developmental stages of ticks at 7–15 d after molting (****p<0.0001). B. Scanning electron microscope images of different tick stages. a. larvae, b. nymph, c. female adult. The scale bars = 50 μm. The orange arrow points to the sensory hair of the anterior pit. C. Response of ticks to cinnamaldehyde repellency after removal of the first pair of legs. These data show the repellent response throughout 120 min, where the repellency percentage declined from 97.33% to 0 at the nymph stage and from 93.75% to 49.67% at the female adult stage. Data are expressed as percentages (*p<0.05, ***p<0.001). D. EAG response of ticks to different concentrations of cinnamaldehyde. From the initial cinnamaldehyde solution to a $10^{-6}$ dilution as −0.035, −0.025, −0.019, −0.018, −0.016, −0.015, −0.013 mV. In contrast, there was no response to paraffin oil (Control group, as solvent) (−0.0073 mV) (p<0.05).

and female adults showed a significant decrease in cinnamaldehyde sensing ability, the repellency percentage declined from 97.33% to 0 at the nymph stage and from 93.75% to 49.67% at the adult stage (*p< 0.05, ***p<0.001) (Fig 1C). In addition, cinnamaldehyde elicited concentration-dependent electroantennagram (EAG) responses at the female adult Haller's

organ. From the initial cinnamaldehyde solution to a $10^{-6}$ dilution as −0.035, −0.025, −0.019, −0.018, −0.016, −0.015, −0.013 mV. In contrast, there was no response to paraffin oil (Control group, as solvent) (−0.0073 mV) ($p < 0.05$) (Fig 1D).

### Characterization of the HL-IR of ticks.

Transcriptomic research revealed differential susceptibility to cinnamaldehyde by unfed ticks at various developmental stages. The number of gene changes was higher at the nymph stage than at the larval and adult stages. Homologs of the ionotropic receptor were significantly altered, revealing the involvement of the olfactory system during the repellent process (S2 Fig). Based on the analysis of transcriptomes after cinnamaldehyde stimulation, we screened a putative ionotropic receptor from cDNA libraries of *H. longicornis* that belonged to the Kainate family of IRs and named this as HL-IR (*H. longicornis* ionotropic receptor) (GenBank accession no. PP999129) (S3 Fig). Phylogenetic analysis of HL-IR indicated a close relationship to various arthropods, particularly ticks (Fig 2A). RT-qPCR indicated that HL-IR was involved in biological processes at multiple developmental stages and tissues. Additionally, HL-IR was expressed in the ganglia and leg at various stages (Fig 2B). To examine its localization in the native state, we subcloned HL-IR into EGFP-N1 and expressed it in human embryonic kidney (HEK) 293T cells. The results showed the HL-IR gene was expressed on the cell membrane (Fig 2C).

### HL-IR is a key molecule for cinnamaldehyde repellence.

We wanted to confirm the role of HL-IR in the repellent reaction of ticks from cinnamaldehyde. HL-IR was tested using qRT-PCR and was highly increased after 30 min of cinnamaldehyde stimulation at different stages (Fig 3A). We examined the transcripts of HL-IR in the first pair of legs. An *in situ* hybridization analysis produced a significant signal on Haller's organ and was apparent in the inner lumen (Fig 3B). Furthermore, we generated HL-IR knockdowns in ticks using RNA interference technology. As expected, when HL-IR was knocked down (Fig 3C), the repellency by cinnamaldehyde was significantly reduced compared to wild-type ticks at the nymphal stage, the repellency rate decreased from 100%, 100%, 100% to 53.33%, 50%, 50% (****$p < 0.0001$) (Fig 3D). Similarly, RNAi of HL-IR significantly reduced the response of Haller's organ to cinnamaldehyde (Fig 3E), The EAG response decreased from −0.0332 mv to −0.0156 mv (****$p < 0.0001$).

### Amino acid 218 is the binding site for HL-IR and cinnamaldehyde.

A binding assay was employed to verify the activity of HL-IR to cinnamaldehyde. A single recombinant band of HL-IR was harvested, and the binding constant of the recombinant HL-IR to 1-NPN was 24.43 μM (Fig 4A). Of the four repellents, only cinnamaldehyde was a successful substitute probe for the HL-IR/1-NPN complex at concentrations up to 40 μM, indicating that cinnamaldehyde was a specific ligand for HL-IR (Fig 4B). The results of HL-IR with cinnamaldehyde constructed by MM-GBSA showed an XP Gscore of −2.743 and MM-GBSA dG Bind of −29.47 kcal/mol, indicating that cinnamaldehyde could stably bind with HL-IR. Structural and molecular docking analyses showed that cinnamaldehyde could penetrate the active pocket of the HL-IR protein, creating π-π interactions with its PHE219 and two hydrogen bonds with ASN218 (Fig 4C). ASN218 was mutated to HIS218, and PHE219 to CYS219 to create a double mutation. Purified mutant proteins were then assayed for competitive binding. The results showed that the cinnamaldehyde binding capacity was significantly decreased after the ASN218 mutation (Fig 4D).

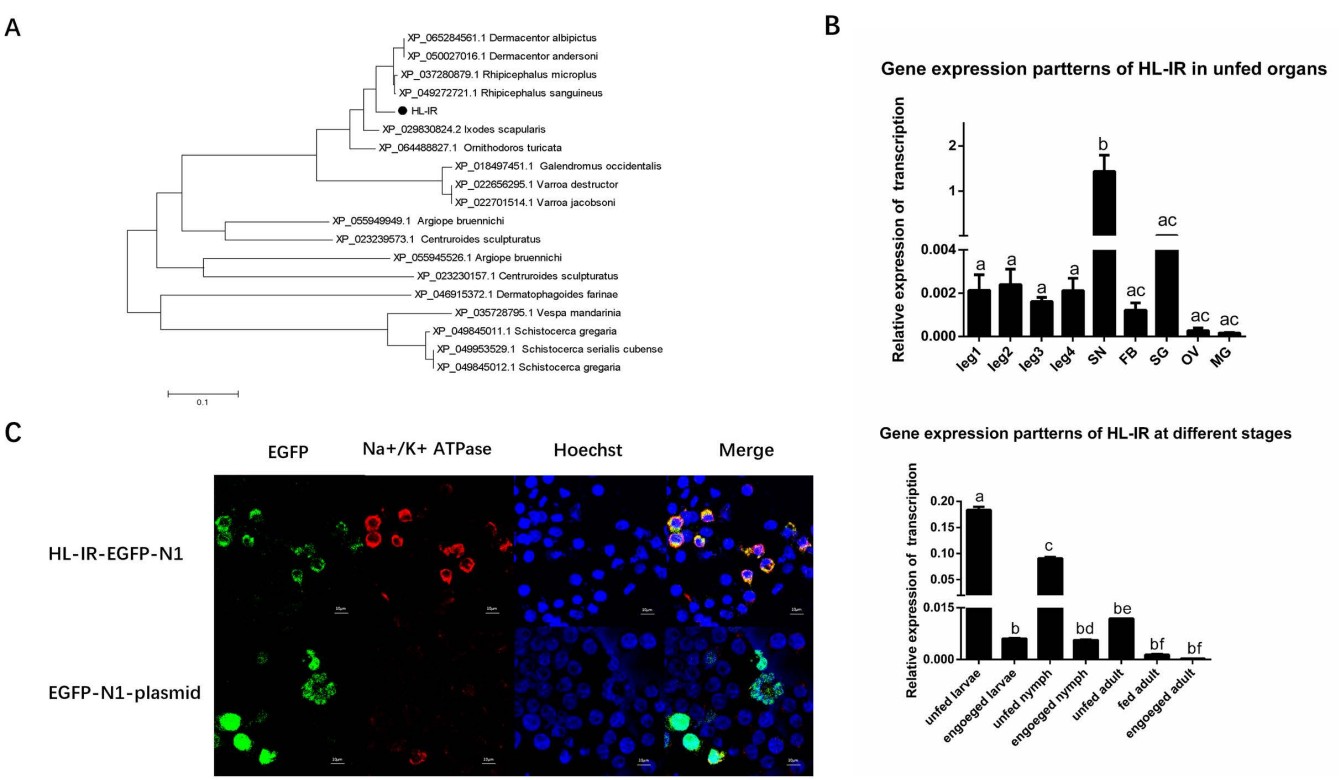

**Fig 2. Identification of HL-IR.** A. Relationships among HL-IR homologs from other species by phylogenetic analysis using maximum likelihood. HL-IR is marked with black circles. B. Transcription analysis of HL-IR during different developmental stages and in different tissues of female adult ticks. SN: ganglion, FB: fat body, SG: salivary glands, OV: ovary, MG: midgut. Data are presented after ANOVA and multiple comparisons (p < 0.05). C. Subcellular localization of HL-IR. Green Fluorescence: the protein containing the EGFP fluorescent tag. Red Fluorescence: Na⁺/K⁺-ATPase (Cell membrane). Blue Fluorescence: Cell nucleus. The scale bars = 10 μm.

## Discussion

Ticks may locate hosts by olfaction in the natural environment [30,41,42]. A repellent response occurs when encountering an unpleasant odor or adverse conditions [43,44]. Transcriptome results have shown that ticks have chemosensory genes that encode proteins involved in repellence behavior. Transcriptomic screening in *H. longicornis* identified HL-IR as a molecular target of natural repellents that showed high sensitivity to cinnamaldehyde, a compound derived from cinnamon essential oil. Indeed, we identified 5 different IRs (other IRs GenBank accession no. PQ741019, PQ741020, PQ741021, PQ741022) in the transcriptome and localized them subcellularly. The results showed that other 4 IRs were not localize to the cell membrane, and we speculated that they might not be able to bind directly to odor molecules (S4 Fig). IRs mediate neuronal communication at synapses in the animal nervous system [45]. In this study, we classified HL-IR into the Kainate family based on sequence domains. IRs comprise several subunits. Typically, there is one specific odorant receptor and one or two common IRcos (IR25a, IR8a, and IR76b) [46–48]. Here, we did not examine whether IRco plays a role in the ability of cinnamaldehyde to repel ticks. Additionally, in the repelling experiment, the nymphs were the most sensitive stage to cinnamaldehyde, a finding that was confirmed by transcriptome data. This phenomenon emphasizes the necessity to investigate nymphs, which was one reason why we used them as experimental material. we speculate that it may be due to the more moderate size of the individual nymph. Nymph

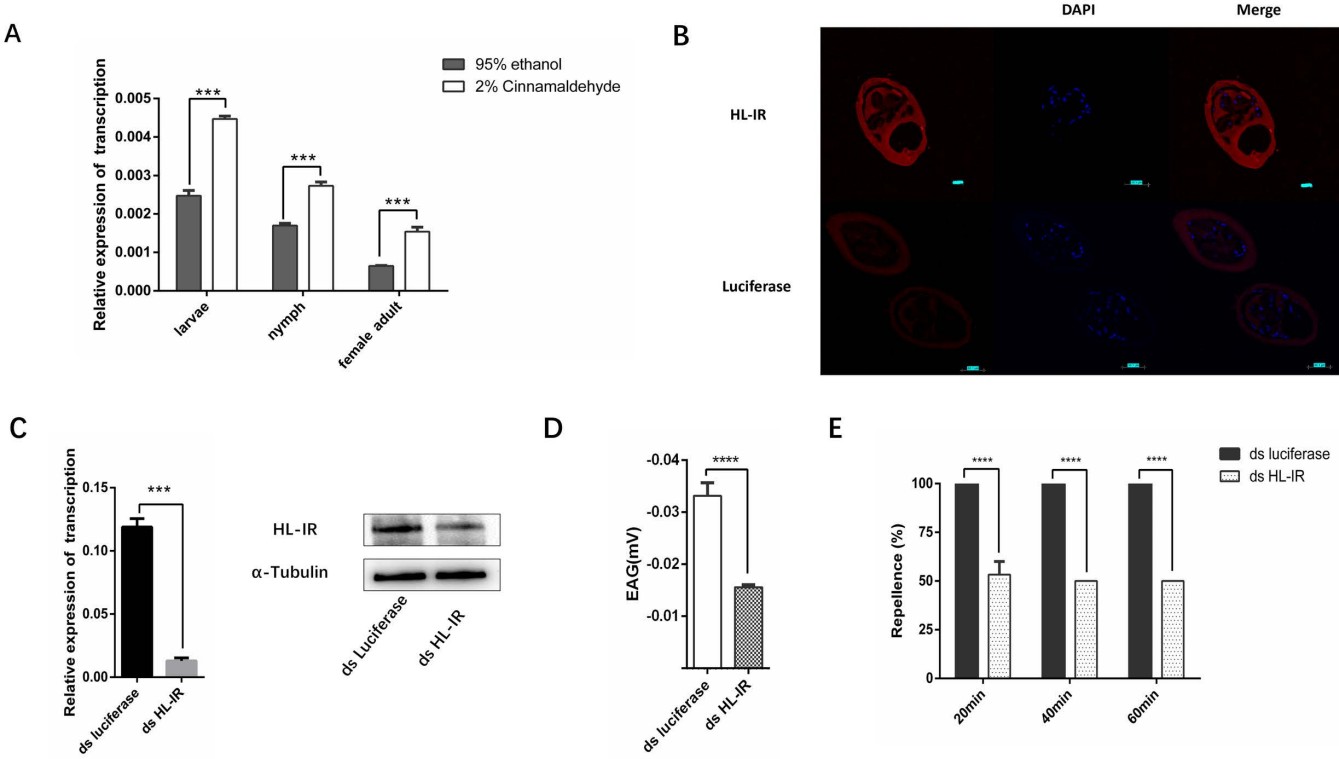

**Fig 3. HL-IR is crucial for ticks to recognize cinnamaldehyde.** A. Determination of HL-IR transcript levels after cinnamaldehyde stimulation (***p < 0.001). B. Expression of HL-IR in Haller's organ by *in situ* hybridization. A synthetic Luciferase probe was used as a control. Red Fluorescence shows HL-IR; Blue Fluorescence shows the cell nucleus. The scale bars = 50 μm. C. Confirmation of RNAi. Transcription level (left) and Expression levels (right) of HL-IR after RNAi. D. Calculation of cinnamaldehyde repellency after RNAi. The repellency rate decreased from 100%, 100%, 100% to 53.33%, 50%, 50% (****p < 0.0001). E. EAG detection of cinnamaldehyde after RNAi. The EAG response decreased from −0.0332 mv to −0.0156 mv (****p < 0.0001).

are more developed than larvae (larvae have three pairs of legs, whereas nymphs and female adults have four pairs), and nymphs are more sensitive than female adults due to their smaller size and lack of environmental adaptation.

IRs can detect taste, odor, volatile acids and amines, humidity, temperature, infrared radiation, and circadian rhythms in insects [37,49–56]. Additional tick species have been found to express IR, and the first leg of *Ixodes scapularis* was shown to express two molecules, IR25a and IR93a, at high levels. These may be connected to the sensory functions of Haller's organ [38]. Phylogenetic analysis revealed that HL-IR was more closely related to arachnids than to insects, indicating potential distinctions in the roles of IRs in different taxa.

Our investigation discovered that Haller's organ was extremely sensitive to cinnamaldehyde. We used the EAG approach to assessing tick responses to volatile compounds (plant odorants and pheromones) [32,57]. Unisensory recordings are frequently utilized to assess smell and taste in insects [58,59]. However, Haller's sensilla are clustered, making it difficult to measure their single sensillum recording (SSR) Naturally, the EAG response may also suggest that the perception of cinnamaldehyde is modulated and responsive to neurofeedback. In insects, IRs are expressed in sensilla, including antennae, wings, legs, and the labellum [60,61]. Similarly, the expression of HL-IR on the leg and in the ganglia implies a role for HL-IR in neurotransmission of chemical sensations. To further investigate the direct relationship between IR and cinnamaldehyde, we attempted to perform whole-cell recordings during cinnamaldehyde stimulation using heterologous expression of HL-IR in HEK 293T cells with

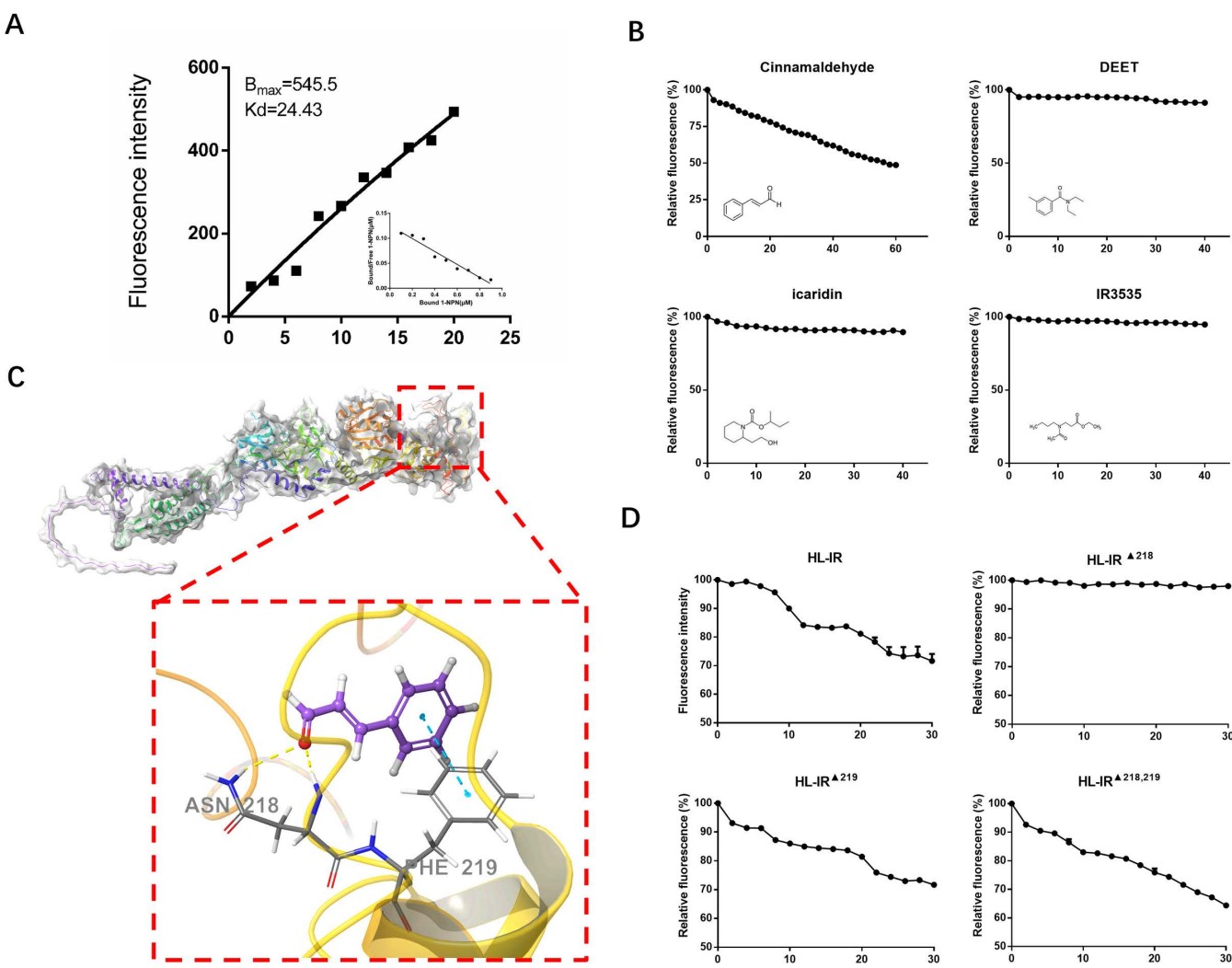

**Fig 4. Cinnamaldehyde is a specific ligand for HL-IR.** A. Binding curve and Scatchard equation for HL-IR. The ratio of bound/free 1-NPN gradually decreased as the concentration of 1-NPN increased, accompanied by a corresponding rise in fluorescence intensity. B. competitive combination curves of four ligands with HL-IR. The figure highlights how the fluorescence intensity decreases with increasing ligand cinnamaldehyde concentration. Fluorescence competitive binding is demonstrated by HL-IR and cinnamaldehyde. C. Molecular docking map (3D) and site prediction of cinnamaldehyde with HL-IR. ASN218 and PHE219 amino acid positions were expected to be the sites of intermolecular interactions with cinnamaldehyde. D. Competitive combination curves of cinnamaldehyde with mutation of HL-IR. The figure illustrates that only the fluorescent competition binding experiment failed following the mutation at 218.

reference to prior research studies in order to discover the interaction between HL-IR and cinnamaldehyde[62]. Regretfully, this experiment was unsuccessful.

Referring to the binding experiments of Odorant binding proteins (OBP) and odorant receptors (OR) in insects [63,64], further research using a variety of ligands, including cinnamaldehyde, revealed that HL-IR has a specific binding capacity of less than 30 μM. The HL-IR ligand molecule was cinnamaldehyde, as demonstrated by competitive binding studies with several repellents. In contrast, HL-IR was insensitive to synthetic chemical repellents such as DEET, IR3535, and icaridin. Although various compounds have repellent effects on ticks, additional olfactory molecules may be involved in mediating the reaction. OBP and odorant receptors linked with plant-derived repellents reported in other arthropods were

not discovered in the transcriptomic data [62,65,66]. Only one of the 43 IRs found by tick genomic research was linked to cinnamaldehyde in our analysis. Thus, further studies on ticks' perception of their surroundings are warranted.

Our findings imply that chemoreceptors and chemosensors may be involved in the repellent process, since knockdown of HL-IR directly decreased the nymphal Haller organ's rejection of cinnamaldehyde and the action of EAG. The percentage of tick repellence decreased from 100% to roughly 50% following HL-IR RNAi, suggesting that HL-IR is not the only mechanism by which ticks react to cinnamaldehyde. Additional research on arthropods revealed that pyrethrum has a dual-targeting mechanism in mosquitoes [67], while *Drosophila*, which are repelled by DEET, displayed multiple channels [68]. Future research should further investigate other processes underlying natural tick repellents.

As a whole, this study found that Haller's organ is sensitive to cinnamaldehyde, and screening for candidate gene families related to sensory input confirmed the requirement of HL-IR in tick cinnamaldehyde repellency. The findings are significant for the development of novel repellent solutions based on cinnamaldehyde that target tick vectors of infectious zoonotic diseases. The results also offer fresh concepts and avenues for further investigation into the molecular characterization of the tick olfactory system.

## Supporting information

**S1 Table.  Primer sequences.**
(XLSX)

**S1 Fig.  Cinnamaldehyde repellency test within 6 hours in parthenogenesis *H. longicornis*.** Data were analyzed by ANOVA, Dunnett's Multiple Comparison-test, and differences are indicated by different letters. A. Cinnamaldehyde repellency test within 6 hours in parthenogenesis larvae. The mean repellency of cinnamaldehyde at concentrations of 0.25%, 0.5%, 1%, 2%, and 4% over a 360min period was 39.12%, 54.90%, 87.56%, 94.03%, and 99.14%, respectively. Notably, the 4% concentration of cinnamaldehyde achieved over 90% repellency at all time points within the 360min duration. In the positive control group, the mean repellency of 20% DEET for the same period was 97.67%. **B. Cinnamaldehyde repellency test within 6 hours in parthenogenesis nymph.** The average repellency of cinnamaldehyde at the 0.25%, 0.5%, 1%, 2%, and 4% concentrations was recorded as 68.22%, 76.06%, 99.36%, 99.32%, and 95.72%, respectively. The 1%, 2%, and 4% concentrations of cinnamaldehyde maintained repellency rates exceeding 90% at all time points during the 360min assay. The mean repellency of 20% DEET during this stage in the positive control group was 95.72%. **C. Cinnamaldehyde repellency test within 6 hours in female adult.** The average repellency rates of cinnamaldehyde at concentrations of 0.25%, 0.5%, 1%, 2%, and 4% over the 6-hour period were 0%, 21.76%, 50.59%, 69.91%, and 98.16%, respectively. The 4% concentration of cinnamaldehyde consistently achieved over 90% repellency at all time points, indicating a significant repellent effect on female adult. The average repellency rate of 20% DEET for 6 hours in the positive control group was 91.84%.
(TIF)

**S2 Fig.  Transcriptome sequencing and differential analysis of *H. longicornis* at different developmental stages in response to cinnamaldehyde stimulation.** A**. Venn analysis of the number of differential transcripts of *H. longicornis* at different developmental stages.** A_: female adult, N_: nymph, L_: larvae, C1: control group, T2: treat group. **B. Number of differentially expressed genes between unstimulated and stimulated ticks in different developmental stages.** Red (upregulated) and yellow (downregulated). Adult: female adult,

Nymph: nymph, Larvae: larvae. **C. Cinnamaldehyde stimulates changes in transcript levels of IR-related genes at different developmental stages.** The bar indicates (l-r) downregulated (red) to upregulated (green) with $-5 < \log2$ normalized fold change $< 5$.
(TIF)

**S3 Fig.  Sequences analysis of HL-IR. A. Conserved domain analysis results for HL-IR. B. Alignment of HL-IR amino acid sequences with those of other species.**
(TIF)

**S4 Fig.  IRs -EGFP-N1 (red) in transfected HEK 293T cells assessed by confocal microscopy.** Green Fluorescence: the protein containing the EGFP fluorescent tag. Red Fluorescence: $Na^+/K^+$-ATPase (Cell membrane). Blue Fluorescence: Cell nucleus. The scale bars = 10 μm.
(TIF)

## Acknowledgments

We thank members of Sibao Wang at CAS Center for Excellence in Molecular Plant Sciences for their help with the EAG assay. We thank LetPub (www.letpub.com) for its linguistic assistance during the preparation of this manuscript.

## Author contributions

**Conceptualization:** Ceyan Kuang, Jinlin Zhou.

**Formal analysis:** Ceyan Kuang, Han Shi, Houshuang Zhang, Yanan Wang, Jinlin Zhou.

**Funding acquisition:** Jinlin Zhou.

**Investigation:** Ceyan Kuang, Han Shi, Jie Cao, Yongzhi Zhou, Jinlin Zhou.

**Methodology:** Ceyan Kuang, Jie Cao, Yongzhi Zhou.

**Supervision:** Houshuang Zhang, Yanan Wang.

**Writing – original draft:** Ceyan Kuang.

**Writing – review & editing:** Jinlin Zhou.

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
