## [Decision Letter · Decision Letter 0]

3 Nov 2024

PNTD-D-24-01490HL-IR mediates cinnamaldehyde repellency behavior in parthenogenetic Haemaphysalis longicornisPLOS Neglected Tropical Diseases Dear Dr. Zhou, Thank you for submitting your manuscript to PLOS Neglected Tropical Diseases. After careful consideration, we feel that it has merit but does not fully meet PLOS Neglected Tropical Diseases's publication criteria as it currently stands. Therefore, we invite you to submit a revised version of the manuscript that addresses the points raised during the review process. Please submit your revised manuscript within 60 days Jan 02 2025 11:59PM. If you will need more time than this to complete your revisions, please reply to this message or contact the journal office at plosntds@plos.org. Please include the following items when submitting your revised manuscript:* A rebuttal letter that responds to each point raised by the editor and reviewer(s). You should upload this letter as a separate file labeled 'Response to Reviewers '. This file does not need to include responses to any formatting updates and technical items listed in the 'Journal Requirements' section below.* A marked-up copy of your manuscript that highlights changes made to the original version. You should upload this as a separate file labeled 'Revised Manuscript with Track Changes '.* An unmarked version of your revised paper without tracked changes. You should upload this as a separate file labeled 'Manuscript '. If you would like to make changes to your financial disclosure, competing interests statement, or data availability statement, please make these updates within the submission form at the time of resubmission. Guidelines for resubmitting your figure files are available below the reviewer comments at the end of this letter. We look forward to receiving your revised manuscript. Kind regards, Clarence Mang'era, PhDGuest EditorPLOS Neglected Tropical Diseases Paul MirejiSection EditorPLOS Neglected Tropical Diseases

Shaden Kamhawi

co-Editor-in-Chief

Paul Brindley

co-Editor-in-Chief

 **Journal Requirements:** **Additional Editor Comments (if provided):** Improve the description and clarity of the methodology section among other corrections as indicated by the reviewers.**Reviewers' Comments:** Reviewer's Responses to Questions

**Key Review Criteria Required for Acceptance?**

**Methods**

-Are the objectives of the study clearly articulated with a clear testable hypothesis stated?

-Is the study design appropriate to address the stated objectives?

-Is the population clearly described and appropriate for the hypothesis being tested?

-Is the sample size sufficient to ensure adequate power to address the hypothesis being tested?

-Were correct statistical analysis used to support conclusions?

-Are there concerns about ethical or regulatory requirements being met?

Reviewer #1: (No Response)

Reviewer #2: No additional experiments are required.

However, materials and methods should be clearly declared.

Reviewer #3: no

**Results**

-Does the analysis presented match the analysis plan?

-Are the results clearly and completely presented?

-Are the figures (Tables, Images) of sufficient quality for clarity?

Reviewer #1: (No Response)

Reviewer #2: Results are sufficient as is. But some additional discussiona is required.

Reviewer #3: no

**Conclusions**

-Are the conclusions supported by the data presented?

-Are the limitations of analysis clearly described?

-Do the authors discuss how these data can be helpful to advance our understanding of the topic under study?

-Is public health relevance addressed?

Reviewer #1: (No Response)

Reviewer #2: The conclusions are well supported by the reuslts.

Reviewer #3: no

**Editorial and Data Presentation Modifications?**

Reviewer #1: (No Response)

Reviewer #2: (No Response)

Reviewer #3: no

**Summary and General Comments**

Reviewer #1: (No Response)

Reviewer #2: This study analyzed the repellent effect of cinnamaldehyde in Haemaphysalis longicornis and its correlation with HL-IR. It is a well-designed experiment and the results are interesting. However, for a clearer analysis of the results, this reviewer suggests the following modifications.

1. The Introduction is missing a lot of information about the purpose and background of the experiment. For example, there is no explanation of why cinnamaldehyde, the chemical used in the test, was chosen. There is only a partial description of Chinese cinnamon oil.

2. The origin of the H. longicornis used in this study should be explained.

3. Many ticks were used in this experiment, and the batch information of the ticks is missing. Are all the ticks from the same batch? If no, need to add description on the batch.

4. The term ‘adult’ is used throughout the paper, including in Fig. 1. But ‘female’ or ‘female adult’ is correct as the strain used is a parthenogenetic strain.

5. Discussion. Lines 348-355 are more appropriately to be included in the Introduction.

6. As results, nymphs were the most sensitive to cinnamaldehyde. A discussion on the reason why, and a comparison of the results with other repellent experiments should be added.

Minor comments

1. line 84. regents -> reagents

2. Line 145. strains -> colonies. Is this correct?

Reviewer #3: Firstly, authors take a result about cinnamaldehyde repellency behavior in tick, they relate this repellency behavior in tick with its Haller's organ, a kind of olfactory sensory structure. Further to analyze a transcriptome data from cinnamaldehyde repellency experiments, they found ionotropic receptor (HL-IR) different expression levels caused the repellency behavior. They characterize this receptor at various. Interfering with HL-IR express level at nymphal stage, they interestingly found that significant decreases in the tick repellency rate from cinnamaldehyde and the response of Haller's organ. To understand the mechanism between HL-IR and cinnamaldehyde repellency behavior, authors with their colleagues validate a interaction effect in above two molecules and explore the critical binding site between the two molecules. Each part of experiment is tightly close. However, certain aspects of the work need to strengthened to ensure all conclusions are well supported.

1) In S1 Fig1C, there is no significant difference between the 0.25% and 0.5% in group of 20min and 40min, but the different labels were signed.

2) In S1 Fig1, please use same type of error bar

3) Couldn't find any structure differences in scanning electron microscope, author reported that larvae lack a sensory hair.

4) Ionotropic receptor (HL-IR) was a core molecule for further mechanism analysis, and HL-IR was obtained form the transcriptomic research. The research process of how important is this molecule in the repellency behavior effects was weak. In addition to this, the data presentation is very sloppy, greatly detracting for the manuscript. a) Fig note can't correspond to fig tilte in S2 FigA. b) The quantity of transcript of nymph was more than 10 times than other groups in S2 FigA and B, which is unreasonable to analyze intergroup differences. c) 9 transcripts were regarded as "IR-relate" genes in S2 FigC, but this result couldn't be supported with any materials. d) A suggestion about plotting, it could be better if you label "IR-relate" genes on the S2 FigB at the same time.

5) line 289-291: " To examine its function in the native state,......". I think author just wanted to detected the localization of HL-IR in cell, neither of its function.

6) A interesting phenomenon is that cinnamaldehyde repellency response were decrease after removal of the first pair of legs (Fig1C) but expression of HL-IR were similar among tick's legs (Fig2B). Wouldn't that suggest that HL-IR in Haller's organ were not a major molecule to receive cinnamaldehyde signals?

7) 2-∆∆Ct method was calculated all RT-PCR experiment data, but I couldn't find in all PR-PCR figure that a given group were normalized to 1. Especially in interferences experiments, the control group should be normalized to 1.

8) Using the competitive fluorescent ligand binding assays to odorant-binding proteins is classic method but limited as tian et al[1] described. Please provide more proof methods.

[1] Tan J, Zaremska V, Lim S, Knoll W, Pelosi P. Probe-dependence of competitive fluorescent ligand binding assays to odorant-binding proteins. Anal Bioanal Chem. 2020 Jan;412(3):547-554.

9) Does "SSR" referring in line 354 mean single sensillum recording. If so, please provide the full name!

10) There are extensive content in second and third paragraphs about Odorant binding proteins, odorant receptors, and gustatory receptor neurons. However, I cannot figure out the correlation between this part and this study. Are the authors trying to demonstrate that HL-IR belongs to one of the three?

PLOS authors have the option to publish the peer review history of their article (what does this mean? ). If published, this will include your full peer review and any attached files.

**Do you want your identity to be public for this peer review?** For information about this choice, including consent withdrawal, please see our Privacy Policy .

Reviewer #1: No

Reviewer #2: No

Reviewer #3: No

---

## [Decision Letter · Decision Letter 1]

28 Jan 2025

Dear Dr. Zhou,

We are pleased to inform you that your manuscript 'HL-IR mediates cinnamaldehyde repellency behavior in parthenogenetic Haemaphysalis longicornis' has been provisionally accepted for publication in PLOS Neglected Tropical Diseases.

Best regards,

Clarence Mang'era, PhD

Guest Editor

Paul Mireji

Section Editor

Shaden Kamhawi

co-Editor-in-Chief

Paul Brindley

co-Editor-in-Chief

Reviewer's Responses to Questions

PLOS authors have the option to publish the peer review history of their article (what does this mean? ). If published, this will include your full peer review and any attached files.

**Do you want your identity to be public for this peer review?** For information about this choice, including consent withdrawal, please see our Privacy Policy .

Reviewer #1: No

Reviewer #2: No

Reviewer #3: No

**Summary and General Comments**

Reviewer #3: After reviewing the manuscript (PNTD-D-24-01490R1), most of the issues have been resolved. However, I still adhere to the following points: 1）In the transcriptome data, the data volume of the nymph group is several times more than that of other groups. I suspect that the sequencing depth of nymph group is inconsistent with that of other groups. 2）When dealing with the data of gene relative expression levels, a normalization process should be carried out.

Nevertheless, the above mentioned issues do not affect the core argument of the article, that is, the experiments designed can demonstrate the mediating role of HL-IR in the repellent behavior of Haemaphysalis longicornis to cinnamaldehyde.

---

## [Editor Report · Acceptance letter]

Dear Dr. Zhou,

We are delighted to inform you that your manuscript, "HL-IR mediates cinnamaldehyde repellency behavior in parthenogenetic Haemaphysalis longicornis," has been formally accepted for publication in PLOS Neglected Tropical Diseases.

Best regards,

Shaden Kamhawi

co-Editor-in-Chief

Paul Brindley

co-Editor-in-Chief
